# Lovastatin fails to improve motor performance and survival in *methyl-CpG-binding protein2*-null mice

Claudia Villani[1], Giuseppina Sacchetti[1], Renzo Bagnati[2], Alice Passoni[2], Federica Fusco[3], Mirjana Carli[1], Roberto William Invernizzi[1]*

[1]Laboratory of Neurochemistry and Behaviour, Department of Neuroscience, Istituto di Ricerche Farmacologiche Mario Negri, Milano, Italy; [2]Analytical Instrumentation Unit, Department of Environmental Health Sciences, Istituto di Ricerche Farmacologiche Mario Negri, Milano, Italy; [3]Genetics of Neurodegenerative Diseases Unit, Department of Neuroscience, Istituto di Ricerche Farmacologiche Mario Negri, Milano, Italy

*For correspondence:
rinvernizzi@marionegri.it

**Competing interests:** The authors declare that no competing interests exist.

**Abstract** Previous studies provided evidence for the alteration of brain cholesterol homeostasis in 129.*Mecp2*-null mice, an experimental model of Rett syndrome. The efficacy of statins in improving motor symptoms and prolonging survival of mutant mice suggested a potential role of statins in the therapy of Rett syndrome. In the present study, we show that *Mecp2* deletion had no effect on brain and reduced serum cholesterol levels and lovastatin (1.5 mg/kg, twice weekly as in the previous study) had no effects on motor deficits and survival when *Mecp2* deletion was expressed on a background strain (C57BL/6J; B6) differing from that used in the earlier study. These findings indicate that the effects of statins may be background specific and raise important issues to consider when contemplating clinical trials. The reduction of the brain cholesterol metabolite 24S-hydroxycholesterol (24S-OHC) found in B6.*Mecp2*-null mice suggests the occurrence of changes in brain cholesterol metabolism and the potential utility of using plasma levels of 24S-OHC as a biomarker of brain cholesterol homeostasis in RTT.

## Introduction

Rett syndrome (RTT), is a rare genetic disease caused by mutations of the X-linked gene methyl-CpG-binding protein2 (*MECP2*), in more than 95% of patients (*Amir et al., 1999*; *Chahrour and Zoghbi, 2007*). Affected subjects, almost exclusively females, show progressive loss of acquired motor and cognitive skills and develop additional symptoms including stereotypic hand movements, drug-resistant seizures, breathing and autonomic dysfunctions (*Chahrour and Zoghbi, 2007*). Increased cholesterol plasma levels and changes in the activity/expression of genes and proteins involved in the cellular uptake, synthesis and feedback regulation of circulating cholesterol have recently been found in RTT individuals carrying *MECP2* mutations (*Grillo et al., 2013*; *Justice et al., 2013*; *Segatto et al., 2014*), However, only one-third of RTT patients show elevated levels of plasma cholesterol (*Justice et al., 2013*).

Cholesterol homeostasis is important for central nervous system function and defects in cholesterol synthesis may cause neurologic symptoms (*Waterham, 2006*). Thus, altered cholesterol homeostasis may have a role in the pathogenesis of RTT and constitute a potential therapeutic target for a condition lacking specific treatments. The relationship between RTT and cholesterol is supported by recent findings showing that inactivation of *Sqle,* encoding squalene epoxidase, a key enzyme in the synthesis of cholesterol (*Pfrieger and Ungerer, 2011*), and pharmacological

treatment with the cholesterol lowering drugs statins improved motor deficit and enhanced survival in *Mecp2*-null mice (*Buchovecky et al., 2013*), a widely used experimental model of RTT (*Ricceri et al., 2008*). *Mecp2*-null mice show complex changes in cholesterol levels, synthesis and metabolism in the brain and periphery, which depend on age, sex and line of mutant, among other factors (*Buchovecky et al., 2013*). Transient increase of total brain cholesterol levels is observed in 56 day (PND56) old male 129.*Mecp2*$^{tm1.1Bird/y}$ mice, but not at PND70, while serum cholesterol increased at both ages in males 129.*Mecp2*$^{tm1.1Bird/y}$ but not in 129.*Mecp2*$^{tm1.1Bird/+}$ heterozygous females or in another line of mutant mice, the B6.Mecp2$^{tm1.1Jae}$ raised on a different background strain (*Buchovecky et al., 2013*). Interestingly, *Cyp46a1*, encoding cholesterol-24-hydroxylase, the neuronal enzyme converting cholesterol into 24S-hydroxycholesterol (24S-OHC), increased at PND28 and markedly decreased in PND56 mutant mice (*Buchovecky et al., 2013*). The reduced expression of *Cyp46a1* was replicated in PND56 B6.*Mecp2*$^{tm1.1Jae}$ mice suggesting that this is a robust and background independent phenomenon that may lead to altered metabolism of cholesterol in RTT. In spite 24S-OHC has emerged as a biomarker of brain cholesterol metabolism in various neurological diseases (*Leoni and Caccia, 2013*) and a potent positive allosteric modulator of N-methyl-D-aspartate (NMDA) receptors (*Paul et al., 2013*) it is not known whether alterations of this metabolite occur in RTT patients and in mouse models of the disease.

Thus, the relevance of these findings for the potential therapeutic effect of statins in RTT and the lack of replication studies of Buchovecky's and colleagues findings prompted us to re-assess the effects of *Mecp2* deletion on brain and serum cholesterol levels, and the ability of lovastatin to prolong survival and rescue motor deficits in *Mecp2*-null mice on a different background strain (B6), available from a commercial source. Finally, we measured brain levels of 24S-OHC and expression of *Cyp46a1* in mice to assess the influence of *Mecp2* on brain cholesterol metabolism.

## Results

### Effect of lovastatin on body weight gain and survival

Body weight of *Mecp2*-null mice at PND 35 was significantly lower than in WT (10.8 ± 0.7 vs. 16.2 ± 0.7 g, p<0.01; Student's t-test). WT mice gained weight over the three weeks treatment reaching 21.4 ± 0.6 g (range, 15.1–25.0) at PND56 (*Figure 1*). An increase in body weight was also observed in *Mecp2*-null mice between PND35 and PND42 reaching a mean body weight of 14 ± 0.5 g (range, 10.0–16.8) at PND42. Body weight remained stable for about one week and then mice lost weight. Although the onset varied across individual, mouse body weight loss in *Mecp2*-null mice started around PND50 and reached 12.9 ± 0.6 g (range, 8.9–16.6) at PND56. Lovastatin had no significant effects on body weight gain in either genotype (*Figure 1*). ANOVA showed a highly significant effect of genotype ($F_{1,54}$ = 206.8, p<0.0001), but no significant effects of treatment ($F_{1,54}$ = 0.07, p=0.80) or genotype x treatment interaction ($F_{1,54}$ = 0.31, p=0.58).

*Figure 2* shows the survival of WT and *Mecp2*-null mice given lovastatin or vehicle. During treatment, 2 out 15 *Mecp2*-null mice given vehicle and 1 out 15 mice given lovastatin died or were euthanized due to excessive body weight loss. The first event occurred at PND54. Median survival evaluated over the whole mice life was 71 days in both vehicle- and lovastatin-treated *Mecp2*-null mice. The Log-rank test show no significant difference between Mecp2-null mice given vehicle and lovastatin ($\chi^2$, 0.023, p=0.88). All WT mice were alive at the end of the study, regardless of the treatment received.

### Rotarod and hanging-wire test

The effect of lovastatin on rotarod performance in *Mecp2*-null and WT mice is shown in *Figure 3*. Latency to fall in WT mice exposed for the first time to the rotarod, steadily increased from trial 1 to 8. One-way ANOVA revealed a highly significant effect of trials in mice given vehicle ($F_{7,105}$ = 7.66, p<0.0001) and lovastatin ($F_{7,91}$ = 5.26, p<0.0001). *Mecp2*-null mice were also able to learn the task as shown by the increased latency to fall from trial 1 to 8. There was a highly significant effect of trials in mice treated with vehicle ($F_{7,98}$ = 3.32, p<0.0032; one-way ANOVA) and lovastatin ($F_{7,98}$ = 5.34, p<0.0001; one-way ANOVA). Latency to fall in *Mecp2*-null mice was significantly lower than in WT ($F_{1,56}$ = 55.7, p<0.0001; two-way ANOVA). Two weeks treatment with lovastatin had no effect on latency to fall in WT mice as revealed by the non significant effect of treatment

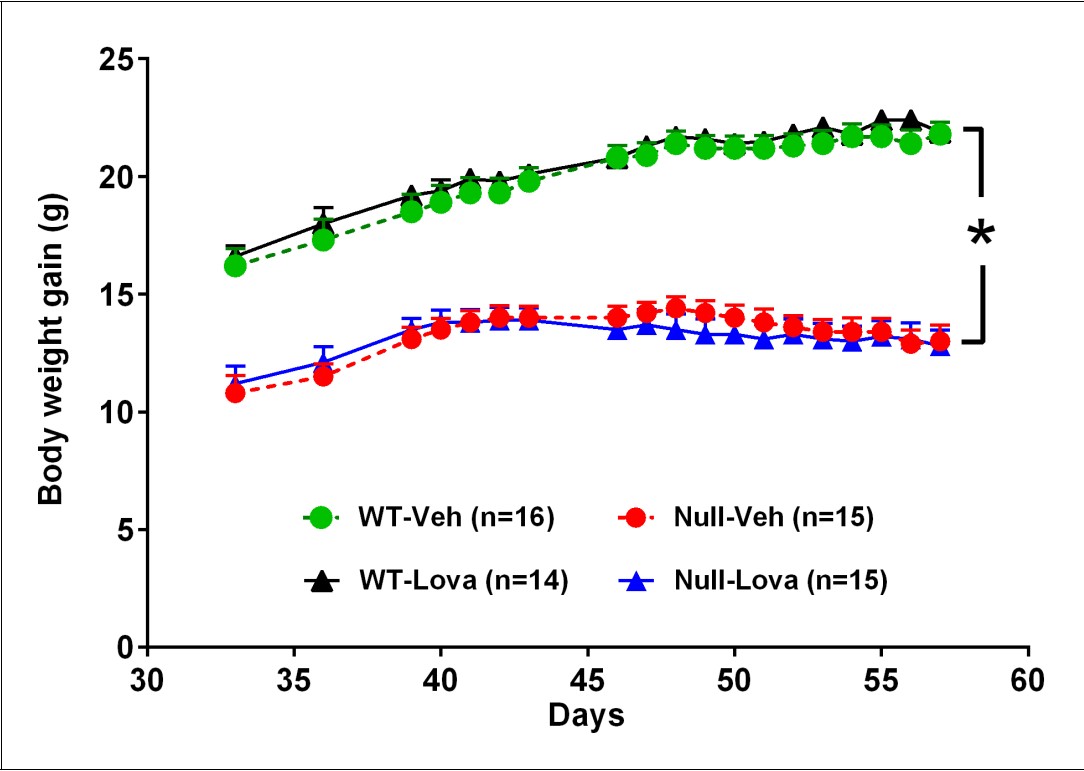

**Figure 1.** Body weight gain in *Mecp2*-null and WT mice and effect of lovastatin. Mice were given vehicle (Veh) or 1.5 mg/kg lovastatin (Lova), twice weekly, from day 35 to 57. Body weight in *Mecp2*-null (Null) mice was significantly lower than in WT. Lovastatin had no significant effects on body weight gain in both genotypes. The number of mice per group is shown in parentheses. *p<0.05 vs. WT (Tukey's test).
The following source data is available for figure 1:

**Source data 1.** Source file for body weight gain.

($F_{1,28}$ = 1.58, p=0.22; two-way ANOVA) and treatment x trial interaction ($F_{7,196}$ = 0.73, p=0.65; two-way ANOVA). Likewise, no significant effects of lovastatin were found in *Mecp2*-null mice (Treatment, $F_{1,28}$ = 0.35, p=0.56; treatment x trial, $F_{7,196}$ = 0.50, p=0.83; two-way ANOVA) (*Figure 3*, left panel).

Treatment with lovastatin was continued for one more week and rotarod performance re-evaluated in the same mice at PND56-57. Although difference in the rotarod performance between genotypes were confirmed ($F_{1,54}$ = 73.9, p<0.0001; two-way ANOVA), lovastatin had no effect in WT (Treatment, $F_{1,28}$ = 0.16, p=0.69; treatment x $trial_{7,196}$ = 0.17, p=0.99; two-way ANOVA) or *Mecp2*-null mice (Treatment, $F_{1,26}$ = 0.12, p=0.74; treatment x trial, $F_{7,182}$ = 0.80, p=0.59, two-way ANOVA) (*Figure 3* right panel).

*Mecp2*-null mice had reduced performance in the hanging-wire test as compared to WT mice (*Figure 4*). Kruskall-Wallis statistic showed highly significant differences at PND50 (29.81; p<0.0001) and PND57 (27.68; p<0.0001). Post-hoc analysis with Dunn's test showed that *Mecp2*-null mice given vehicle had a significantly lower score than WT mice either at PND 50 or 57. Lovastatin had no effect on hanging-wire score either in WT or in *Mecp2*-null mice at any times after treatment (*Figure 4*).

## Levels of cholesterol and 24S-OHC and Cyp46a1 expression

*Table 1* shows the effect of *Mecp2* gene deletion on serum and brain cholesterol levels in 28 and 56 days old mice. Deletion of *Mecp2* significantly reduced serum cholesterol regardless of the age while no differences were found in brain cholesterol content between *Mecp2*-null and WT mice.

24S-OHC was decreased by 10% in the brain of PND56 *Mecp2*-null mice and by 12% in PND177 *Mecp2*[+/-] females (*Table 2*). Although the reduction was small, the effects were highly significant

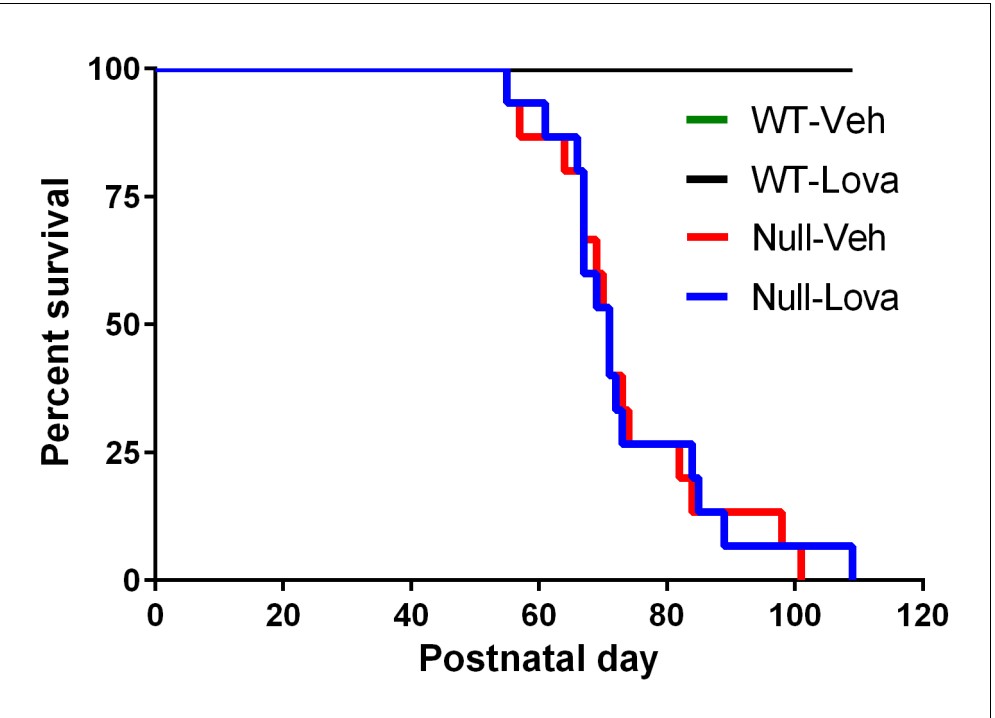

**Figure 2.** Survival curves. Kaplan-Meier plot showing premature mortality in *Mecp2*-null (Null) and WT mice given vehicle (Veh) or lovastatin (Lova). Median survival was 71 days in both vehicle- and lovastatin-treated mice. No mortality was seen in WT mice given either vehicle or lovastatin.

because of the very low variability within groups (Males *Mecp2*-null vs. WT, p=0.0045; females *Mecp2*[+/-] vs., WT p=0.0003; Student's t-test). No significant changes were found in brain 24S-OHC in PND28 *Mecp2*-null male mice.

No significant differences in the expression of brain *Cyp46a1* were observed between *Mecp2*-null male mice and WT either at PND28 or PND56 (*Figure 5*).

## Discussion

Using the *Mecp2*[tm1.1Bird/y] mice, one of the most used mouse models of RTT, we addressed the relevance of the recently proposed role of cholesterol and statins in the pathogenesis and therapy of RTT (*Buchovecky et al., 2013*). We confirmed the robustness and reliability of the model, which in our hands, closely reproduces the originally observed differences in body weight, reduced lifespan and motor deficits between *Mecp2*-null and WT male mice (*Guy et al., 2001*, *2007*; *Ricceri et al., 2008*). However, we were not able to reproduce the raise of brain and serum cholesterol and the beneficial effect of lovastatin on motor performance and survival observed previously in *Mecp2*[tm1.1Bird/y] mice on a different background strain (*Buchovecky et al., 2013*). This limits the generalizability of previous findings on the potential pathogenic role of cholesterol and the efficacy of statins in RTT.

The type of *Mecp2* mutation, dose of lovastatin, route and schedule of drug administration, mouse age, housing, behavioral tests and the lack of cholesterol in the diet used in our study are identical to those used by Buchovecky et al. Thus, other factors may account for the discrepancies between the present and the previous study. The main difference is the background strain of the *Mecp2*[tm1.1Bird/y] mice, B6 in our study and 129 S6/SvEvTac (129) in Buchovecky et al. We observed a reduction of serum cholesterol in PND28 and PND56 *Mecp2*-null mice and no effect on brain cholesterol, regardless the age. The same allele deletion on the 129 background determined the opposite effect on serum and a transient increase of brain cholesterol (*Buchovecky et al., 2013*). It is known that genetic background is a source of variability that influences the expression of phenotype and

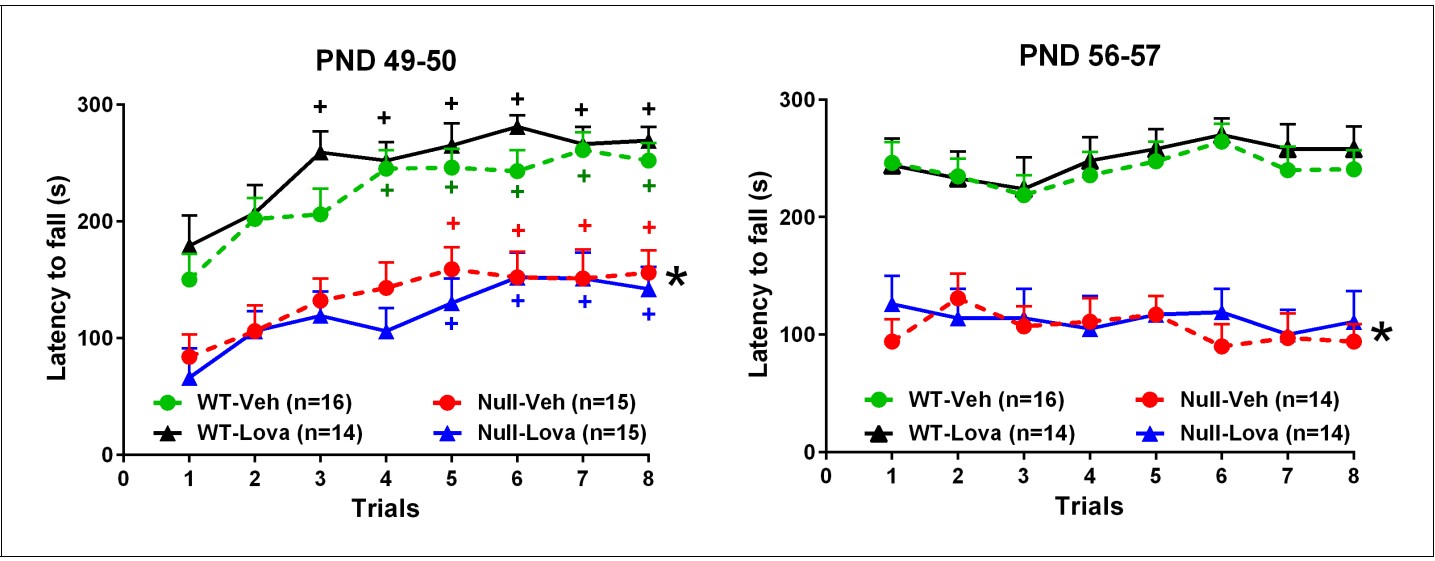

**Figure 3.** Rotarod. Effect of lovastatin (Lova) and vehicle (Veh) in *Mecp2*-null (Null) and WT mice in the rotarod test. Starting on PND 35, mice were given Lova (1.5 mg/kg s.c) or Veh twice weekly and tested on the rotarod 24 hr (trials 1–4) and 48 hr (trials 5–8) after the last dose of the chronic schedule. The test was repeated one week later (PND 56 and 57). The number of mice in each group is shown in parentheses. Mutant mice displayed reduced motor performance compared to WT (*p<0.05 vs. WT mice, Tukey's test), but improved their performance over repeated trials ([+]P < 0.05 vs. trial 1; Tukey's test). Lovastatin had no significant effects on latency to fall at PND 49–50 and 56–57 in both genotypes.

The following source data is available for figure 3:

**Source data 1.** Source file for latency to fall in the rotarod test.

drug response (*Cervo et al., 2005*; *Crawley et al., 1997*; *Sittig et al., 2016*). In mouse models of RTT, genetic background influences the effects of *Mecp2* deletion on body weight, cognitive, anxiety and social phenotypes resulting, in several cases, in directionally opposite effects of the same allele (*Guy et al., 2001*; *Katz et al., 2012*; *Samaco et al., 2013*). *Mecp2*[tm1.1Bird/y] male mice on a 129 background are obese, likely because of a systemic metabolic disorder (*Kyle et al., 2016*), while they are underweight in a B6 background (*Guy et al., 2001*; present study). However, other phenotypes, such as motor deficits, are reproduced in different mouse models of RTT regardless of the genetic background (*Lombardi et al., 2015*; *Samaco et al., 2013*). In agreement with our findings, brain cholesterol and the expression of *Cyp46a1* was not affected in *Mecp2*[tm1.1Bird/y] mice on the B6 background, aged 7 to 56 days and no changes of brain levels of the cholesterol precursor desmosterol were observed at 7 and 14 days (*Lopez et al., 2017*). In addition, no increase in serum and brain cholesterol was detected in another line of *Mecp2*-null mice, the *Mecp2*[tm1.1Jae] on a B6 background (*Buchovecky et al., 2013*). These findings indicate that *Mecp2* deletion is not sufficient by itself to affect brain cholesterol homeostasis but may require the interaction with modifier genes, as already suggested for the influence of genetic background on the effect of *Mecp2* on body weight (*Guy et al., 2001*).

The efficacy of lovastatin in improving motor performance in male 129.*Mecp2*-null mice (*Buchovecky et al., 2013*) cannot be confirmed in mice carrying the same allele on a B6 background. Interestingly, lovastatin had no effects on serum cholesterol in WT mice regardless of the background strain (*Buchovecky et al., 2013*). These findings are consistent with the scarce effect of statins in reducing circulating cholesterol in mice (*Pecoraro et al., 2014*) and indicate that neither the background strain nor the mutation of *Mecp2* alone are sufficient to explain the effects of lovastatin on serum cholesterol observed in 129.*Mecp2*-null mice. This suggests that indirect mechanisms are likely involved in the ability of lovastatin to improve motor performance and survival in *Mecp2*-null mice.

It could be argued that lovastatin may be effective only in mice showing altered cholesterol homeostasis, such as 129.*Mecp2*-null mice. The empirical finding that statins are more effective in

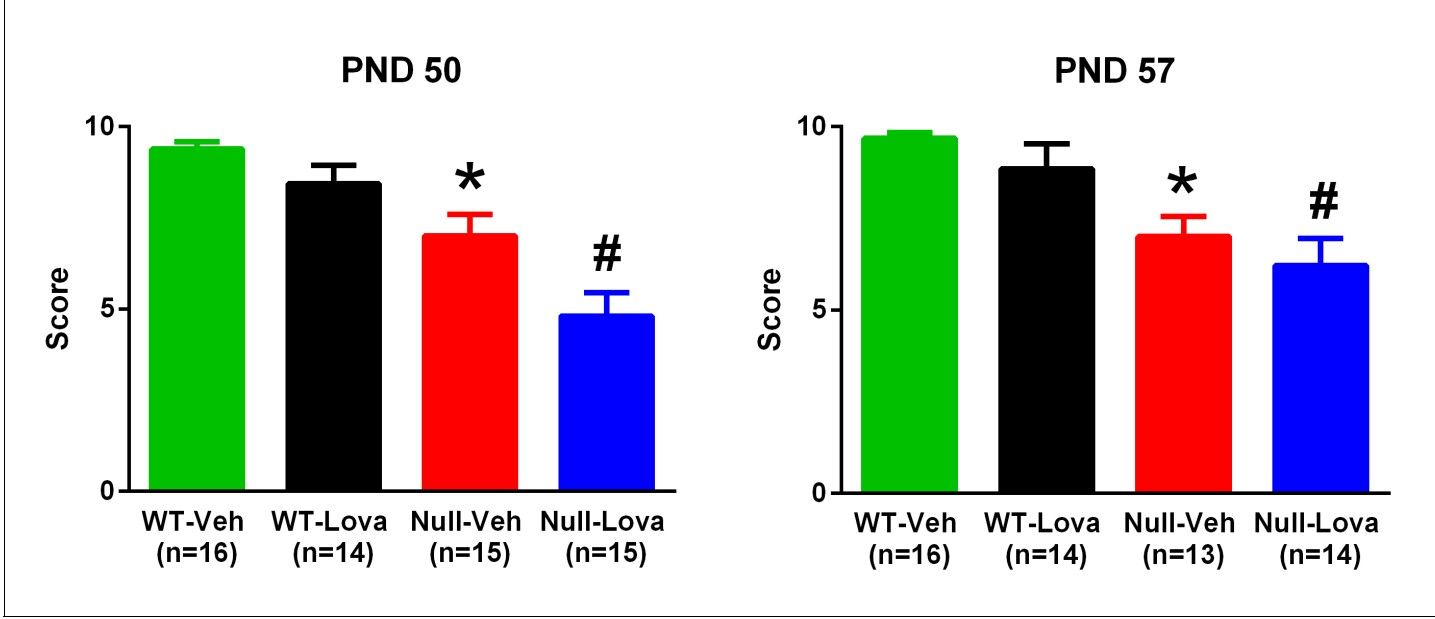

**Figure 4.** Hanging-wire test. Thirty-five days old *Mecp2*-null (Null) and WT mice were given 1.5 mg/kg lovastatin (Lova) and Vehicle (Veh) twice weekly and their performance in the hanging-wire test was assessed on PND 50 and 57. The mice were the same as those used for the rotarod test. The number of mice in each group is shown in parentheses. *p=0.006 vs. WT-Veh, #p=0.0007 vs. WT-Lova (Dunn's test). Lovastatin had no significant effects on the score at PND 49–50 and 56–57 in either genotype.

The following source data is available for figure 4:

**Source data 1.** Source file for hanging-wire scores.

lowering serum cholesterol when this was increased through a cholesterol-enriched diet or because of the deletion of the Apo-E carrier (*Pecoraro et al., 2014*) apparently support this view. However, the fact that fluvastatin is effective in improving motor deficits in 129.*Mecp2*<sup>tm1.1Bird/+</sup> females, in which serum cholesterol levels are not changed, argues against this interpretation. In addition, it is unlikely that the concentration of lovastatin reaching the brain after the administration of 1.5 mg/kg is sufficient to suppress cholesterol synthesis. Even in mice given 3 mg/kg, brain levels of lovastatin were below 25 nM (*Chen et al., 2007*), far less than the $IC_{50}$ and Ki for the inhibition of 3-hydroxy-3-methylglutaryl-CoA (HMG-CoA) reductase estimated in vitro (77 and 150 nM, respectively) (*Bischoff et al., 1997*; *Bischoff and Heller, 1998*). Taken together, these findings suggest that

**Table 1.** Effect of *Mecp2* deletion on serum and brain cholesterol levels.
Serum and brain levels of total cholesterol were expressed as mg/dL and mg/g, respectively. Cholesterol levels were measured in 28 (PND28) and 56 (PND56) day old *Mecp2*-null and WT mice. Food was removed from cage 6 hr before sacrifice. Data are means ± SEM of 5 mice per group. *p<0.05, **p<0.01 vs. WT (Student's t-test).

| Genotype | Serum | | Brain | |
|---|---|---|---|---|
| | PND 28 | PND 56 | PND 28 | PND 56 |
| WT | 99.2 ± 6.8 | 90.8 ± 4.8 | 9.1 ± 0.3 | 10.9 ± 0.5 |
| *Mecp2*-null | 81.7 ± 2.0* | 66.5 ± 5.4** | 9.1 ± 0.5 | 10.3 ± 0.5 |

Source data 1. Source file for serum and brain levels of cholesterol. The Source file contains the concentrations of serum (mg/dL) and brain (mg/g) cholesterol in individual wild type and Mecp2-null mice at PND 28 and PND56.

**Table 2.** Effect of *Mecp2* deletion on brain levels of 24S-OHC.

Brain levels of 24S-OHC, expressed as ng/mg of tissue, were measured in 28 (PND28) and 56 (PND56) day old *Mecp2*-null and WT male mice and in PND177 *Mecp2*$^{+/-}$ females and WT mice. Food was removed from cage 6 hr before sacrifice. Data are means ± SEM of 5–6 mice per group. **p<0.01, ***p<0.001 vs. WT (Student's t-test).

| Genotype | Males | | Females |
|---|---|---|---|
| | PND28 | PND56 | PND177 |
| WT | 37.5 ± 0.8 | 38.4 ± 0.8 | 48.9 ± 0.7 |
| *Mecp2* mutant | 36.1 ± 0.4 | 34.6 ± 0.6** | 43.0 ± 0.6*** |

Source data 1. Source file for brain levels of 24S-hydroxycholesterol. The Source file contains the concentrations of brain (ng/g) 24S-OHC in individual wild type and Mecp2-null male mice on PND28 and PND56 and WT and *Mecp2*$^{+/-}$ females on PND177. Each mice is identified by the ear tag number.

direct inhibition of HMG-CoA reductase is unlikely the mechanism by which lovastatin lowers cholesterol synthesis and improves motor deficits and survival in 129.*Mecp2*-null mice.

Differences in the rotarod protocol between studies deserve some comments. We administered two rotarod sessions to mice on PND49-50 and PND56-57 while in the previous study mice underwent a single session. We were able to show deficits in rotarod performance in our *Mecp2*-null mice comparable to those seen previously in the same line of mutant mice on a B6 background (*Mellios et al., 2014*). Our mice learn the task as testified by the increased performance (latency to fall) from trial 1 to 8 while the performance of untreated 129.*Mecp2*-null mice used in Buchovecky's study was very poor and did not improve across trials. It is unlikely that our protocol prevented the effect of lovastatin in the rotaroad since *Mecp2*-null mice of the same mutant line, genetic background, age, sex, and similar basal performance in the rotarod improved in response to clenbuterol (*Mellios et al., 2014*). In addition, using another motor test, the hanging-wire, we confirmed that

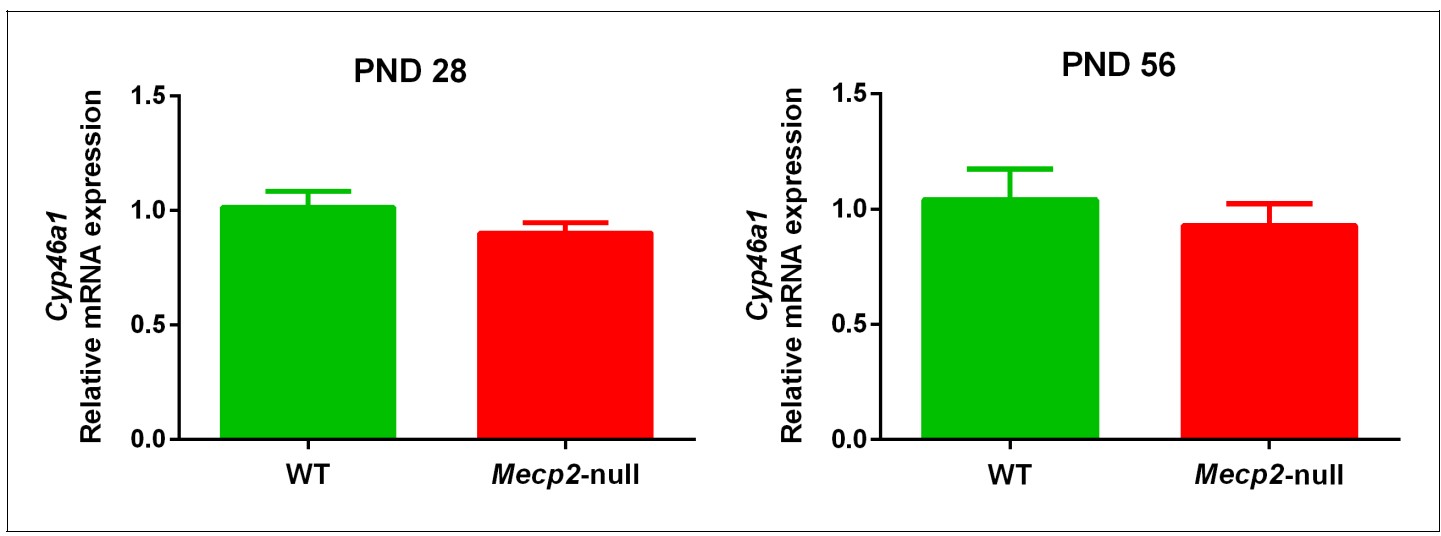

**Figure 5.** Cyp46a1 expression.  mRNA expression of *Cyp46a1* in the brain of 28 (PND28) and 56 (PND56) day old *Mecp2*-null and WT male mice. Expression levels were relative to respective WT. Data are means ± SEM of 7 mice per group. No significant differences were observed between WT and Mecp2-null mice at PND28 (p=0.21) and PND56 (p=0.51) (Student's t-test).

The following source data is available for figure 5:

**Source data 1.** Source file for brain mRNA expression of *Cyp46a1*.

lovastatin failed to rescue motor deficits of mutant mice. Thus, it is unlikely that our protocol masked potential effects of lovastatin on motor performance. However, we cannot exclude that previous learning of the task precluded the assessment of any effects lovastatin could have on motor learning in PND56-57 mice.

We found no changes in the expression of brain *Cyp46a1* in PND28 and PND56 *Mecp2*-null mice. This fully agrees with recent data showing that *Cyp46a1* expression was not affected in B6.*Mecp2*-null mice aged 14, 43 or 56 days (*Lopez et al., 2017*), but is in sharp contrast with previous findings (*Buchovecky et al., 2013*). However, we found a moderate, but highly significant reduction of brain 24S-OHC in males and confirmed this effect in females *Mecp2$^{+/-}$* mice. Further studies are needed to clarify whether the reduction of 24S-OHC may be attributable to reduced activity of cholesterol-24-hydroxylase or other mechanisms. Since 24S-OHC is produced to a large extent by brain neurons (*Björkhem et al., 1998*), low levels suggests reduced metabolism of neuronal cholesterol. The fact that 24S-OHC was reduced in fully symptomatic mutant mice, but not in younger, pre-symptomatic mice suggests that reduced cholesterol metabolism is mainly associated to the symptomatic stage of the disease. Interestingly, 24S-OHC was recently found to control neuron excitability through allosteric modulation of the NMDA receptor (*Paul et al., 2013*). Excitatory/inhibitory imbalance is found in the brain of *Mecp2* mutant mice. Selective impairment in excitatory synaptic transmission as well as elevation of neuronal activity occurs in different brain regions of *Mecp2*-null mice (*Calfa et al., 2015*; *Dani et al., 2005*; *Nelson et al., 2006*; *Shepherd and Katz, 2011*). Altered composition of NMDA receptor subunits in cortical circuits contribute to the regression of visual cortical function in *Mecp2*-null mice (*Durand et al., 2012*) and the NMDA receptor antagonist ketamine has been shown to reverse functional deficits of forebrain circuits' activity induced by *Mecp2* deletion (*Kron et al., 2012*). These findings support the involvement of NMDA receptors in the excitatory/inhibitory imbalance found in the brain of *Mecp2* mutant mice and suggest that changes in brain levels of 24S-OHC, by modulating NMDA receptor activity may contribute to Mecp2-induced alterations of brain circuits' excitability.

The importance of replication studies has recently been highlighted by the failure to confirm the potential benefit of bone marrow transplantation in a mouse model of RTT (*Derecki et al., 2012*; *Wang et al., 2015*). Assessing the influence of background strain on the phenotypic expression of Mecp2 mutation may ultimately contribute to our understanding of the causes of the disease, the mechanism involved in drug action and the development of treatments for patients with Rett syndrome.

The present results show that lovastatin had no effect on motor performance and survival when the deletion of the Mecp2 allele is expressed on the particular background strain we have studied and do not negate the role of cholesterol and the potential therapeutic effect of statins in RTT. The present findings suggest that the response to lovastatin is under the control of inherent genetic or other biological mechanisms specific to the effects of lovastatin on brain function. Thus, the efficacy of lovastatin in RTT may be limited to those patients that reproduce alteration of cholesterol homeostasis through mechanisms similar to those described in mice responding to statins (*Buchovecky et al., 2013*). Further studies are needed to establish the mechanisms involved in the reduction of brain levels of 24S-OHC, its role in RTT and the feasibility of using plasma levels of this metabolite as a biomarker of brain cholesterol metabolism, which may help to sort out patients that may best respond to treatment aimed at re-balancing cholesterol homeostasis.

## Materials and methods

### Mice breeding and genotyping

Males, B6.*Mecp2*$^{tm1.1Bird/y}$ were used in all experiments except for measurement of 24S-OHC where *Mecp2$^{+/-}$* females were also tested. *Mecp2$^{+/-}$* and wild type (WT) mice were purchased from The Jackson Laboratories and maintained in a specific pathogen free animal facility under standard conditions of temperature (21 ± 2°C) and humidity (55 ± 5%), with food (Teklad Global 2018S; Envigo, Italy) and water *ad libitum*. Mice to be used in the experiments were generated at our facility by mating female *Mecp2$^{+/-}$* to WT male mice, both on a pure C57BL/6J (B6) background strain, in 2:1 female to male ratio per box and genotyped using standard procedures. Mice were group housed

and provided with environmental enrichment consisting of a coloured plastic refugee in each home cage. Nesting material was provided to breeding cages.

## Experimental design

The timeline of drug treatment and behavioural tests is shown in *Figure 6*. According to recommended guidelines (*Katz et al., 2012*; *Kilkenny et al., 2010*), 13–16 mice per group were used in behavioural studies. Four cohorts of mice of the same age at the beginning of treatment (PND 35 ± 2) and testing (PND 49–50 and 56–57), balanced for genotype and treatment were tested during a three months period and data were pooled. Mice were randomly allocated to experimental groups using computer generated randomization schedules (www.statpages.org) and researchers conducting the experiments, specifically behavioural assessment and drug treatment, were blinded to the experimental groups. The Istituto di Ricerche Farmacologiche 'Mario Negri' adheres to the principle set out in the following law, regulations, and policies governing the care and use of laboratory animals: Italian Governing Law (D.lgs.26/2014; Authorisation n. 19/2008 A issued March 6, 2008 by Ministry of Health); Mario Negri Institutional Regulations and Policies providing internal authorization for persons conducting animal experiments (Quality Management System Certificate – UNI EN ISO 9001:2008 – Reg. N° 6121); the NIH Guide for the Care and Use of Laboratory Animals (2011 edition) and EU directives and guidelines (EEC Council Directive 2010/63/UE). The statement of Compliance (Assurance) with the Public Health Service (PHS) Policy on Human Care and Use of Laboratory Animals has been recently reviewed (9/9/2014) and will expire on September 30, 2019 (Animal Welfare Assurance #A5023-01).

## Body weight gain and survival

WT and *Mecp2*-null mice receiving lovastatin or vehicle were regularly weighed before treatment, twice weekly during the first week, five days a week starting from day 39 and daily from day 46 to 56.

In *Mecp2*-null mice symptoms progress rapidly and in the final part of their life sudden, unanticipated death did occur in some subjects while others were euthanized with an overdose of anaesthetic (150 mg/kg ketamine plus 2 mg/kg medetomidine) when weight-loss exceeded 20% for at least 72 hr. The age of death or sacrifice in days was recorded and entered in the survival analysis.

## Behavioural studies

All testing occurred during the light phase of the light-dark cycle. For behavioural studies, test cohorts were composed of *Mecp2*-null mice and respective WT littermates' controls. A total of 30

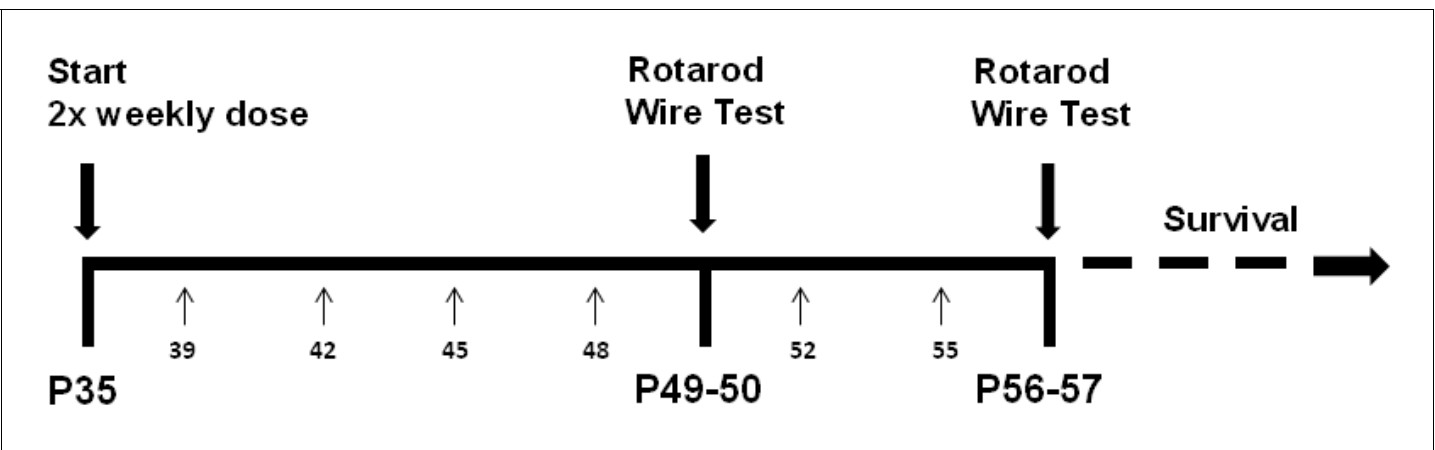

**Figure 6.** Lovastatin injection and behavioural testing schedule. Treatment with 1.5 mg/kg lovastatin or vehicle subcutaneously started at postnatal day 35 (P35) and was repeated every 3–4 days. Small arrows and numbers below the horizontal line indicate the exact postnatal day. Rotarod was done on day 49 (trials 1–4) and 50 (trials 5–8) and repeated on days 56 and 57 (morning). Hanging-wire test was administered on day 50 and repeated on day 57 (afternoon). Last lovastatin injection was on day 55.

WT and 30 *Mecp2*-null mice were used to assess the ability of lovastatin to counteract motor deficits using tests sensitive to *Mecp2* deletion such as accelerating rotarod and hanging-wire (*Buchovecky et al., 2013*; *Stearns et al., 2007*). To reduce the number of animals used, the same subjects were used in both motor tasks at postnatal day (PND) 49–50 and 56–57 according to the scheme shown in *Figure 1*. Fifteen min before testing mice were placed in the testing room for acclimation.

## Rotarod

The mice rotarod apparatus (Ugo Basile, Italy) consisted of rods (5.7 cm long, 3 cm in diameter) suitably machined to provide grip. Five lanes divided by flanges allowed the simultaneous assessment of the motor performance of 5 mice. Height to fall was 16 cm. Each mouse was positioned in a separate lane on a rod rotating at 4 rpm. Once all mice were in place, acceleration started and reached 40 rpm in 300 s. The trial ended when the mouse fell off the rod or after 300 s. The activation of the fall sensor stopped the timer and the latency to fall shown on the display was recorded. When the mouse gripped the rod firmly for 10 s, the trial was ended and the time spent walking on the rod recorded. The procedure was repeated for a total of 4 trials a day for two consecutive days. At the end of each trial, mice were returned to the home cage for 30 min resting before the next trial.

### Hanging-wire

A stainless steel wire 2 mm thick and 640 mm long was suspended horizontally to two vertical stands at 30 cm height above the ground. A mouse was handled by the tail, suspended by the forelimbs on the middle of the wire and the timer started. When the animal reached one of the platforms positioned at the ends of the wire the timer was stopped and a second trial was started by hanging the mouse on the middle of the wire. If the animal fell, the timer was stopped, the initial score (10) was diminished by one and the procedure was restarted. The test lasted for 180 s or for a maximum of 10 falls. Falling scores were collected for each mouse. Wire test was conducted in the afternoon, 6 hr after the last rotarod trial.

## Cholesterol assay

On the day of sacrifice, food was removed from the home cage at least 6 hr before blood collection. Mice were anesthetized with 100 mg/kg ketamine and 1 mg/kg medetomidine IP and sacrificed by decapitation between 2:00 and 3:00 PM. Blood was collected through thoracic wound in 1.5 mL Eppendorf tubes (Eppendorf Srl, Hamburg, Germany) that were stored at RT for 30–60 min. Serum was transferred to clean tubes and frozen at −70°C until analysis.

The brain was extracted from the skull, immediately frozen on dry ice and stored at −70°C. Tissue was homogenized by sonication in 20 volumes reaction buffer consisting of 0.1M potassium phosphate pH 7.4, 0.05M NaCl, 5 mM sodium cholate and 0.1% Triton X-100.

Homogenate was centrifuged for 10 min at 14,000 rpm in an Avanti J-E series centrifuge equipped with a JA 20.1 rotor (Beckman Coulter, Milan, Italy) at 4°C and supernatant stored at −70°C until analysis. Levels of cholesterol in serum and supernatant were determined in duplicate for each sample with the Amplex Red Cholesterol Assay Kit (Thermo Fisher Scientific; Italy) according to manufacturer instructions.

## 24S-OHC and Cyp46a1 assay

### Sample preparation and analysis of 24S-OHC

After 6 hr food-deprivation, mice were anesthetized with 100 mg/kg ketamine and 1 mg/kg medetomidine IP and sacrificed by decapitation between 2:00 and 3:00 PM. Brain samples (200 mg) were homogenized in 2 mL of ethanol/water 4/1 (v/v), containing 1 μg of internal standards (see Chemicals and reagents) and 100 μg of dibutylhydroxytoluene as antioxidant agent. Samples were centrifuged at room temperature for 10 min at 6000 rcf and the supernatants were transferred to glass vessels with pressure tight caps. A saponification procedure was then performed to hydrolyze 24S-OHC ester, by adding 50 μL of 10M potassium hydroxide to each sample and heating for 2 hr at 100°C. After cooling, 2 mL of water and 2 mL of dichloromethane were added to supernatants and vortexed for 2 min. Samples were centrifuged for 10 min at 600 rcf, and the lower organic phase

was separated and dried under nitrogen. Samples were re-suspended in 200 µL of chromatographic mobile phase for instrumental analysis.

HPLC-MS detection of 24S-OHC was performed using an Agilent 1200 Series HPLC system with binary pumps and refrigerated autosampler (+4°C), coupled to a Sciex API5500 triple quadruple mass spectrometer, equipped with a Turbo Ion Spray source.

HPLC separation was performed on an Ascentis Express C18 column, 150 × 2.1 mm, 2.7 µm particle size (Sigma Aldrich, St. Louis, MO), maintained at 37°C. A dual eluent system, consisting of ammonium acetate 5 mM in ultrapure water (A) and ammonium acetate 5 mM in methanol (B), was pumped at a flow rate 160 µL/min. The elution gradient was as follows: 0 min (80% B), 20 min (88% B), 28 min (100% B), 38 min (100% B), 39 min (80% B). The injection volume was 8 µL.

The Turbo Ion Spray settings were: ion spray voltage 5500 V; curtain gas 20 (arbitrary units); collision gas 7 (arbitrary units); source temperature 400°C; nebulizer gas 40 and desolvating gas 30 (arbitrary units). Mass spectrometric analyses were performed with positive ions in the SRM mode (Selected Reaction Monitoring), using the two most abundant fragmentation products of the ammonium adduct ions of each analyte. The selected transitions, together with optimized instrumental conditions, are listed in *Supplementary file 1*.

## Cyp46a1 expression

Brain RNA was isolated using SV Total RNA Isolation System (Promega Corporation, Madison, USA), according to manufacturer's instructions. First strand complementary DNA (cDNA) was synthesized from 1000 ng of total RNA using High Capacity cDNA Reverse Transcription Kits (Applied Biosystems) according to manufacturer's instructions. RT-PCR was performed in triplicate for each sample on 7300 Real Time PCR System (Applied Biosystems CA). Reactions contained 100 ng of cDNA, 1 µL Taqman probe containing forward and reverse primers, 10 µL TaqMan Universal Master Mix II and water to a final volume of 20 µL. PCR conditions were as follows: 50°C for 2 min, 95°C for 10 min, 40 cycles of 95°C for 15 s and 60°C for 60 s. Gene expression was normalized to an *RpL19* (*L19*) internal loading control and analyzed using the $2^{-(\Delta\Delta CT)}$ method expressed to relative WT. Gene primers for QRT-PCR were: for *Cyp46a1*, forward primer (5'-TCCTCTCCTGTTCAGCACCT-3') and reverse primer (5'-GGCCATGACAACTTTCACCT-3'); for *RpL19*, forward primer (5'-TTCCCGAGTACAG-CACCTTTGAC-3') and reverse primer (5'CACGGCTTTGGCTTCATTTTAAC-3').

## Drug treatment

Lovastatin was obtained by Teva Pharmaceutical Works Co. Ltd. (Debrecen, Hungary) and solution prepared as described elsewhere (*Kita et al., 1980*). Briefly, 12 mg lovastatin were dissolved in 1.8 mL 100% ethanol and 0.9 mL NaOH (0.6 M) and 18 mL sterile water were mixed to lovastatin solution and left at room temperature. After 30 min 3.5 mL of 10x PBS (Thermo Fisher Scientific, Milan, Italy) were added and the pH of the solution adjusted to 7.2 with 0.6 M HCl. The final volume was adjusted to 40 mL with sterile water. Vehicle was prepared in the same way excluding lovastatin. Drug solution was prepared once weekly, stored at 2–8°C and warmed to RT before administration. Stability of lovastatin solution was guaranteed by HPLC analysis showing no significant degradation after three weeks storage at 2–8°C (data not shown). Lovastatin was dosed at 1.5 mg/kg subcutaneously twice weekly for three weeks starting at PND35 as described in *Figure 1*. Behavioral tests were done 24–48 hr after the last drug administration.

## Chemicals and reagents

All solvents were HPLC/MS grade. Methanol was from Fluka (Buchs, Switzerland), acetonitrile was from Sigma-Aldrich (Steinheim, Germany). $K_2HPO_4 \cdot 3H_2O$, $KH_2PO_4$ and NaCl were from Merck (Darmstadt, Germany), sodium cholate and Triton X-100 from Sigma-Aldrich (Steinheim, Germany), and ammonium acetate from Carlo Erba (Milan, Italy). Water for eluents and sample preparation was of HPLC grade, prepared in-house (Nanopure Diamond, Barnstead, Switzerland and MILLI-RO PLUS 90, MILLIPORE, Molsheim, France). Standards of 22(R)-hydroxycholesterol, 22(S)-hydroxycholesterol and 24(R,S)-hydroxycholesterol-D7 were from Avanti Polar Lipids (Spectra2000, Rome, Italy). Standards of 24(R)-hydroxycholesterol, 24(S)-hydroxycholesterol, 25-hydroxycholesterol, 25-hydroxycholesterol-D6, 25(R)−27-hydroxycholesterol and 25(S)−27-hydroxycholesterol were from Cayman Chemical (Vinci Biochem, Vinci, Italy).

## Statistics

Kaplan-Meier curves and statistics (Log-rank) were used for survival data representation and analysis (*Kaplan and Meier, 1958*). All other data were expressed as mean ± SEM. The body weight gain and latency to fall in the rotarod test were analyzed by repeated measures ANOVA with genotype (*Mecp2*-null and WT) and treatment (lovastatin and vehicle) as between-subject factors and days or trials as within-subject factor. Further analysis with repeated measures ANOVA were conducted to assess the improvement of latency to fall across trials in mice given vehicle. Post-hoc comparisons between means were done by Tukey's test. The effects of lovastatin and vehicle in *Mecp2*-null and WT mice in the hanging wire test were compared by Kruskall-Wallis test, followed by Dunn's test corrected for multiple comparisons. Serum and brain levels of cholesterol, brain 24S-OHC and *Cyp46a1* expression in *Mecp2*-null and WT mice were compared by Student's t-test. Survival analysis and non-parametric tests was carried out on a PC with GraphPad Prism 6.07 (GraphPad Software, Inc., USA). All other statistics were performed by StatView 5.0.1 (SAS Institute, Inc., USA).

## Acknowledgements

We thank Dr D Albani for his expertise and advices in developing the RT-PCR method for *Cyp46a1*. The Laboratory of Neurochemistry and Behaviour of the Istituto di Ricerche Farmacologiche 'Mario Negri' is member of the AIRETT Research Team promoted by the Associazione Italiana Sindrome di Rett (AIRETT). We thank Teva Pharmaceutical Works Co Ltd (Hungary) for the generous gift of lovastatin.

## Additional information

### Funding

| Funder | Grant reference number | Author |
| --- | --- | --- |
| Istituto di Ricerche Farmacolo-giche Mario Negri | Intramural funding | Roberto William Invernizzi |

The funders had no role in study design, data collection and interpretation, or the decision to submit the work for publication.

### Author contributions

CV, GS, Acquisition of data, Analysis and interpretation of data, Drafting or revising the article; RB, AP, Acquisition of data, Drafting or revising the article; FF, Acquisition of data, Analysis and interpretation of data; MC, RWI, Conception and design, Acquisition of data, Analysis and interpretation of data, Drafting or revising the article

### Author ORCIDs

Claudia Villani, http://orcid.org/0000-0001-6334-9013
Roberto William Invernizzi, http://orcid.org/0000-0002-6017-9781

### Ethics

Animal experimentation: The Istituto di Ricerche Farmacologiche "Mario Negri" adheres to the principle set out in the following law, regulations, and policies governing the care and use of laboratory animals: Italian Governing Law (D.lgs.26/2014; Authorisation n. 19/2008-A issued March 6, 2008 by Ministry of Health); Mario Negri Institutional Regulations and Policies providing internal authorization for persons conducting animal experiments (Quality Management System Certificate - UNI EN ISO 9001:2008 - Reg. N° 6121); the NIH Guide for the Care and Use of Laboratory Animals (2011 edition) and EU directives and guidelines (EEC Council Directive 2010/63/UE). The statement of Compliance (Assurance) with the Public Health Service (PHS) Policy on Human Care and Use of Laboratory Animals has been recently reviewed (9/9/2014) and will expire on September 30, 2019 (Animal Welfare Assurance #A5023-01). The protocol was approved by the Italian Ministry of Health (Permit Number 946/2015-PR).

## Additional files

**Supplementary files**
• Supplementary file 1. Instrumental conditions for the HPLC-MS analysis of hydroxysterols.

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
