## [Decision Letter]

Congratulations, we are pleased to inform you that your article, "Lovastatin fails to improve motor performance and survival in *methyl-CpG-binding protein2*-null mice", has been accepted for publication in *eLife*. If you have selected our "Publish on Acceptance" option, your PDF will be published within a few days; if you have opted out of the "Publish on Acceptance" option, your work will be published in about four weeks' time.

Please take note of the points below and we hope you will continue to support *eLife* going forward. We ask that you address some of the comments raised below and most importantly make it clear that your data show that there is no effect of Lovastatin in the particular background strain you studied, which is different from the strain used by Buchovecky et al. You can mention that this highlights the fact the effects of statins may be background specific raising important issues to consider when contemplating trials. Something along these lines would be helpful. We look forward to the revised final version.

*Reviewer #1:*

The manuscript by Villani and colleagues examines cholesterol levels in the brain of *Mecp2* null mice on a C57Bl/6J (B6) background and the ability of lovastatin to effect survival and motor performance. The study builds on previous work showing alterations in cholesterol homeostasis in the brain of *Mecp2* null mice on a 129S6/SvEvTac genetic background and the ability of lovastatin to increase survival and motor performance in these mice. The current study presents data that the *Mecp2* null B6 mice do not have altered cholesterol levels in brain, only a slight decrease in serum cholesterol levels, and slightly lower levels of 24S-hydroxycholesterol in brain. The authors also show lovastatin does not rescue survival or motor performance deficits. The authors attribute the difference in cholesterol levels and the response to lovastatin between the studies to the genetic background strain of the mice. The study is well conducted and the findings are interesting as they show that lovastatin may have limited benefits in correcting behavioral phenotypes in mutant mouse models with implications towards the clinic. Given the importance of replicating findings that have clinical implications in the development of treatments for patients with Rett syndrome the study is significant.

*Reviewer #2:*

This is a very well done study demonstrating that Lovastatin does not generally improve the phenotypes of mice lacking Mecp2. This is in contrast to a previous report that indicated that loss of function mutations in *Sqle* would modify in a positive fashion the phenotypes of *Mecp2* mutant mice, and that treatment of these same mice with Lovastatin improved the phenotypes. Critically, the authors of this manuscript used the identical mouse mutation and the same dosing method as was used in the previous manuscript, and they abide by recommendations for rigor in preclinical studies (randomization, blinding, etc.). The major difference between their work and the previous work is the different strain background, with the previous work using 129 and this using B6. This is a very important aspect, and a strength of this study. *Mecp2* mutant mice in a 129 strain are obese and develop a metabolic phenotype, whereas they are underweight in a B6 background. As the authors carefully describe, the potential benefit seen in the previous study was likely due to a strain specific effect, not a general effect of the *Mecp2* mutation. This is very important with regards to the development of new therapies in Rett syndrome, because it raises concerns that statin treatments likely will not be of broad benefit in Rett syndrome. Replication studies such as this are very important to the field and unfortunately are not published enough. It might be worthwhile for the authors to mention recent studies that showed a failure to replicate the benefit of bone marrow transplantation in Rett syndrome (PMID: 25993969), in support of the broad importance of replication studies. In summary, this is an excellent manuscript that deserves to be published in *eLife*.

*Reviewer #3:*

The ability of lovastatin to improve symptoms associated with mutation of the *Mecp2* gene in mice is examined. The data show that administration of lovastatin at 1.5 mg/kg body weight to male knockout mice for a two or three week period failed to improve performance in rotarod or hanging-wire tests and did not extend lifespan. Lipid measurements reveal reduced serum cholesterol levels in the knockout mice but no changes in bran cholesterol levels. Slightly lower levels of 24-hydroxycholesterol are observed in the brains of older male and female mutant mice but levels of the enzyme responsible for this metabolite, cholesterol 24-hydroxylase, are not different between control and knockout animals.

These findings are in contrast to an earlier paper published in 2013 by Justice and colleagues, which reported broad changes in sterol metabolism in *Mecp2*-deficient mice and an ability of statins to ameliorate many of the Rett-syndrome-like symptoms associated with loss of this enzyme.

As the earlier studies were done in mice with an inbred 129svEvTac background and the present studies were done with the same mutation on a Black6 background, the authors conclude that the beneficial effects of statins in this animal model system are strain-specific.

There are several shortcomings associated with the present study. First, that the background on which the *Mecp2* mutation was maintained had a large effect on statin outcomes was reported by Justice and colleagues in 2013. The present study confirms these findings but does not extend them. Second, only two lipids (cholesterol and 24-hydroxycholesterol) were examined here, whereas the lipid analyses reported in the Justice paper were more extensive. Third, since control studies were not performed here in a known responsive strain (SvEv), it is not possible to determine whether the administered statin was consumed, absorbed, or otherwise effective in the Black6 mice.

---

## [Author Response]

*[…] We ask that you address some of the comments raised below and most importantly make it clear that your data show that there is no effect of Lovastatin in the particular background strain you studied, which is different from the strain used by Buchovecky et al. You can mention that this highlights the fact the effects of statins may be background specific raising important issues to consider when contemplating trials. Something along these lines would be helpful. We look forward to the revised final version.*

*Reviewer #1:*

*The manuscript by Villani and colleagues examines cholesterol levels in the brain of Mecp2 null mice on a C57Bl/6J (B6) background and the ability of lovastatin to effect survival and motor performance. The study builds on previous work showing alterations in cholesterol homeostasis in the brain of Mecp2 null mice on a 129S6/SvEvTac genetic background and the ability of lovastatin to increase survival and motor performance in these mice. The current study presents data that the Mecp2 null B6 mice do not have altered cholesterol levels in brain, only a slight decrease in serum cholesterol levels, and slightly lower levels of 24S-hydroxycholesterol in brain. The authors also show lovastatin does not rescue survival or motor performance deficits. The authors attribute the difference in cholesterol levels and the response to lovastatin between the studies to the genetic background strain of the mice. The study is well conducted and the findings are interesting as they show that lovastatin may have limited benefits in correcting behavioral phenotypes in mutant mouse models with implications towards the clinic. Given the importance of replicating findings that have clinical implications in the development of treatments for patients with Rett syndrome the study is significant.*

*Reviewer #2:*

*This is a very well done study demonstrating that Lovastatin does not generally improve the phenotypes of mice lacking Mecp2. This is in contrast to a previous report that indicated that loss of function mutations in Sqle would modify in a positive fashion the phenotypes of Mecp2 mutant mice, and that treatment of these same mice with Lovastatin improved the phenotypes. Critically, the authors of this manuscript used the identical mouse mutation and the same dosing method as was used in the previous manuscript, and they abide by recommendations for rigor in preclinical studies (randomization, blinding, etc.). The major difference between their work and the previous work is the different strain background, with the previous work using 129 and this using B6. This is a very important aspect, and a strength of this study. Mecp2 mutant mice in a 129 strain are obese and develop a metabolic phenotype, whereas they are underweight in a B6 background. As the authors carefully describe, the potential benefit seen in the previous study was likely due to a strain specific effect, not a general effect of the Mecp2 mutation. This is very important with regards to the development of new therapies in Rett syndrome, because it raises concerns that statin treatments likely will not be of broad benefit in Rett syndrome. Replication studies such as this are very important to the field and unfortunately are not published enough. It might be worthwhile for the authors to mention recent studies that showed a failure to replicate the benefit of bone marrow transplantation in Rett syndrome (PMID: 25993969), in support of the broad importance of replication studies. In summary, this is an excellent manuscript that deserves to be published in eLife.*

That there is no effect of Lovastatin in the particular mouse strain we have studied is now made clear in the Abstract and at the end of the Discussion.

The failure to replicate the beneficial effects of bone marrow transplantation in mice is now mentioned and briefly commented at the end of the Discussion to highlight the importance of replication studies.

*Reviewer #3:*

*[…] There are several shortcomings associated with the present study. First, that the background on which the Mecp2 mutation was maintained had a large effect on statin outcomes was reported by Justice and colleagues in 2013. The present study confirms these findings but does not extend them.*

We would like to point out that Justice’s study compared the effects of Mecp2 deletion on cholesterol homeostasis in two lines of mutants, the 129.*Mecp2^tm1.1Bird/y^* B6.Mecp2^tm1.1Jae^ male mice, which differ for both the type of mutation and the background strain. However, the effects of statins on cholesterol levels, survival and motor performance were only studied in a single strain, the 129.*Mecp2^tm1.1Bird^*. In consideration of the above, we think that our study extends the findings reported in Justice’s article.

*Second, only two lipids (cholesterol and 24-hydroxycholesterol) were examined here, whereas the lipid analyses reported in the Justice paper were more extensive.*

The main focus of our study was to re-evaluate the potential efficacy of a statin in reversing the motor deficits and prolonging survival for the obvious reason that the efficacy of statins in the Mecp2-null experimental model of RTT suggests a therapeutic option for the treatment of Rett patients that may be readily tested in clinical trials. This and the limited funding available suggested to us to focus our investigations on a few key points. After we submitted our manuscript, a new paper confirmed the failure of *Mecp2* deletion to affect brain cholesterol and *Cyp46a1* expression in the same line of B6.*Mecp2*-null mice used in our study. In addition, the paper provided a more complete lipid analysis, which improved our knowledge on the effects of Mecp2 deletion on lipid metabolism. This article is now mentioned in the revised manuscript (Discussion section).

*Third, since control studies were not performed here in a known responsive strain (SvEv), it is not possible to determine whether the administered statin was consumed, absorbed, or otherwise effective in the Black6 mice.*

We agree that the direct comparison with a “responsive strain” would have increased the impact of our findings. With regards to the second part of the comment, we would like to point out that lovastatin was administered subcutaneously, hence, it did not need to be consumed. Available evidence suggest that lovastatin is absorbed and reach the brain of B6 mice better than less hydrophobic statins, such as pravastatin (Johnson-Anuna et al. 2005, JPET 312:786) and it effectively prevented audiogenic seizures in an experimental model of fragile X syndrome in B6 mice (Osterweil et al. 2013, Neuron 77:243).